# Remanence Increase in SrFe_12_O_19_/Fe Exchange-Decoupled Hard-Soft Composite Magnets Owing to Dipolar Interactions

**DOI:** 10.3390/nano13142097

**Published:** 2023-07-18

**Authors:** Jesús Carlos Guzmán-Mínguez, Cecilia Granados-Miralles, Patrick Kuntschke, César de Julián Fernández, Sergey Erokhin, Dmitry Berkov, Thomas Schliesch, Jose Francisco Fernández, Adrián Quesada

**Affiliations:** 1Electroceramic Department, Instituto de Cerámica y Vidrio, CSIC, 28049 Madrid, Spain; jesus.guzman@icv.csic.es (J.C.G.-M.); c.granados.miralles@icv.csic.es (C.G.-M.);; 2Max Baermann GmbH, 51429 Bergisch Gladbach, Germanyt.schliesch@max-baermann.de (T.S.); 3Istituto dei Materiali per l’Elettronica e il Magnetismo, CNR, 43124 Parma, Italy; cesar.dejulian@imem.cnr.it; 4General Numerics Research Lab, 07743 Jena, Germany

**Keywords:** hard-soft composites, permanent magnets, ferrites, dipolar interactions

## Abstract

In the search for improved permanent magnets, fueled by the geostrategic and environmental issues associated with rare-earth-based magnets, magnetically hard (high anisotropy)-soft (high magnetization) composite magnets hold promise as alternative magnets that could replace modern permanent magnets, such as rare-earth-based and ceramic magnets, in certain applications. However, so far, the magnetic properties reported for hard-soft composites have been underwhelming. Here, an attempt to further understand the correlation between magnetic and microstructural properties in strontium ferrite-based composites, hard SrFe_12_O_19_ (SFO) ceramics with different contents of Fe particles as soft phase, both in powder and in dense injection molded magnets, is presented. In addition, the influence of soft phase particle dimension, in the nano- and micron-sized regimes, on these properties is studied. While Fe and SFO are not exchange-coupled in our magnets, a remanence that is higher than expected is measured. In fact, in composite injection molded anisotropic (magnetically oriented) magnets, remanence is improved by 2.4% with respect to a pure ferrite identical magnet. The analysis of the experimental results in combination with micromagnetic simulations allows us to establish that the type of interaction between hard and soft phases is of a dipolar nature, and is responsible for the alignment of a fraction of the soft spins with the magnetization of the hard. The mechanism unraveled in this work has implications for the development of novel hard-soft permanent magnets.

## 1. Introduction

The challenge of constantly improving the performance of permanent magnets has been ongoing for decades. Lately, the geostrategic and environmental issues associated with rare-earth elements (REE) as raw materials have brought forward the need to develop alternative permanent magnets that could, at least partially, substitute them [1]. One of the strategies to obtain REE-free substitutes is to fabricate composite magnets, based on a hard magnetic phase with high coercivity and a soft phase with high magnetization, in which the magnetic coupling between hard and soft phases leads to an increase in remanence and energy product, which is one of the figures of merit of permanent magnets [2,3,4,5,6].

The most pursued strategy consists of exploiting the exchange-coupling between hard and soft particles in order to gain remanence while avoiding a detrimental coercivity loss [7,8]. According to the model, provided that the soft particles are below a given size threshold [9] and that structural coherency at the interface exists, effective exchange coupling should align at remanence the spins of the soft with the magnetization of the hard at reasonable coercivity penalties. However, we recently demonstrated that not only is it extremely challenging to meet the structural requirements associated with this approach, but in addition, robust exchange-coupling may lead to a collapse of the coercivity [10,11,12], as a consequence of the lowered onset for domain wall propagation [13].

Focusing on systems based on hexaferrites, such as SrFe_12_O_19_, as the hard magnetic phase, results so far have not met the expectations for exchange-coupled hard-soft composite magnets [14]. In fact, more than 30 years after the pioneering work by Kneller and Hawig [15], the effect is not being exploited by the industry. When using Fe as the soft phase, works in the field have reported increases in magnetization without a significant loss of coercivity [16,17,18,19] in exchange-coupled powders. Unfortunately, the samples studied were neither densified nor magnetically oriented, which does not enable extracting conclusions on the real remanence of the composites. It is important to explain as well, that although single-phase behavior in the demagnetization curves is usually associated with effective exchange-coupling between hard-soft particles, experiments have shown that the coercive field distribution associated with broad particle size distributions in the composite may lead to single-phase curves in the absence of exchange-coupling [11,12].

More recently, the possibility of exploiting dipolar interactions within the composite magnets to improve their remanence, instead of exchange-coupling, has been discussed [10]. In fact, composites consisting of SFO and FeCo nanowires (in which the dipolar interaction, through shape anisotropy, plays a crucial role) have been reported to present a significant increase in remanence and energy product [20].

Here, the microstructural and magnetic properties of magnetically oriented SFO/Fe composite powders and the corresponding anisotropic dense magnets are studied as a function of soft phase content and particle size. The magnetic orientation assures the maximum remanence as required for magnets. Scanning Electron Microscopy (SEM) is employed to reveal particle sizes and geometric distribution of hard and soft phases, while demagnetization curves measured with Vibrating Sample Magnetometry (VSM) and Permagraph instruments detail the magnetic properties of the magnetically aligned powders and magnets. Analyzed in combination with micromagnetic simulations, the correlation between microstructure and magnetic properties will be established.

## 2. Materials and Methods

The raw materials used in this study were commercial strontium ferrite (SrFe_12_O_19_, 99.99%) powders supplied by Max Baermann GmbH (Germany-China) (Bergisch Gladbach, Germany) and various commercial Fe particles (99.99%) with different particle sizes ranging from 50 nm to 11 µm. The hard-soft composite samples compositions were prepared by incorporating 5, 10 and 15 vol% of soft phase to SrFe_12_O_19_ (SFO) powders, using different sizes of Fe particles. All samples were studied both in the form of oriented powders and in the form of injection molded magnets. The preparation of the magnetically oriented powder samples for the magnetic characterization consisted of dispersing 30–50 mg of powder in a bonding glass inside a capsule under an externally applied magnetic field H = 0.3 T generated by a NdFeB N42 magnet. The particles, although in proximity, were isolated from one another after this procedure. The injection molded oriented magnets were fabricated at the company Max Baermann GmbH following their usual industrial production method. The injection-molded pieces were squares of 2 cm in lateral size and 4 mm in height. In order for the demagnetizing boundary conditions to be comparable, all magnet samples had the same sizes and shapes.

The mixing was carried out by means of a low-energy dry mixing method [21]. The dry dispersion process consisted of shaking SrFe_12_O_19_/Fe particle mixtures in a 60 cm^3^ nylon container for 5 min at 50 rpm using a tubular-type mixer (Mixer/Mill, 8000D, SPEX Sample Prep., Metujen, NJ, USA). The particle size and morphology of the powders were evaluated using secondary electron images of field emission scanning electron microscopy, FE-SEM (Hitachi S-4700, Hitachi, Tokyo, Japan). Structural analysis was performed by X-ray diffraction (XRD) with a Bruker D8 diffractometer using Cu Kα radiation (λ = 1.5418 Å) and a Lynxeye XE-T detector. Subsequent Rietveld analysis of the XRD data was performed by FullProf Suite [22]. Thermogravimetric analysis (TGA) in air was employed to determine the oxidation temperature of the Fe powders between RT and 900 °C, using a TA Instruments Q50 system. The magnetic properties of the samples were measured using the custom-made VSM described in reference [23], applying a maximum H-field of 1.4 T, and a Permagraph Instrument (Permagraph C, Magnet-Physik, Cologne, Germany). The measurements were carried out at room temperature, and the sensitivity of the instrument was 0.1 Am^2^/kg and 0.025 mT, according to specifications.

For the micromagnetic simulations, we employed a micromagnetic algorithm initially developed for the simulation of the magnetization distribution of magnetic nanocomposites [24,25]. The magnetization distribution in an isolated sphere of soft phase with diameter D under the influence of an external magnetic field (produced by the hard phase) is simulated by considering the four standard contributions to the total magnetic energy: external field, magnetic anisotropy, exchange and dipolar interaction. The following material parameters (typical for Fe) were used for simulating the soft material: saturation magnetization Ms = 220 Am^2^/kg, cubic magnetocrystalline anisotropy Kcub = 20 kJ/m^3^ with easy axis along the <100> directions and the exchange stiffness constant A^bulk^ = 21 pJ/m [10]. The particles under study had D = 15–50 nm and were discretized in the non-regular mesh with the typical mesh element size of 3 nm. Open boundary conditions were applied in all simulations.

## 3. Results and Discussion

### 3.1. Hard-Soft Composite Powders

Figure 1 shows the SEM micrographs corresponding to the starting SFO powder and the Fe powders with three different particle sizes. The particle distribution in the SFO powders is, as previously reported [26], of a bimodal nature, with larger particles in the 2–5 µm range surrounded by smaller particles in the 200–500 nm range. All SFO particles have platelet shapes, as expected. The SEM characterization of the three Fe powders reveals that the smaller-sized powder, Figure 1b, is formed by particles with an average size of 50 nm that are arranged in contact with each other, presumably for electrostatic and magnetostatic reasons. The powder depicted in Figure 1c again reveals a bimodal particle size distribution, where 100–200 nm Fe particles coexist with larger 1–2 µm particles. For simplicity, we refer to this powder in the following as 1 µm Fe powder. The larger-sized Fe powder, shown in Figure 1d, presents an average particle size of 11 µm.

As the samples were fabricated and manipulated in air conditions, the XRD patterns of the Fe powders were measured in order to study their oxidation state. Figure 2a shows the pattern and refinement of the 50 nm Fe powders as a selected example. From the refinement of the patterns for each Fe powder, we concluded that, for 50 nm Fe particles, the powders were composed of 77 wt% Fe, 17 wt% FeO and 6 wt% Fe_3_O_4_. For 1 µm Fe powders, the sample consisted of 79 wt% Fe, 15 wt% FeO and 6 wt% Fe_3_O_4_. For 11 µm particles, the XRD refinement detected 100 wt% Fe. Figure 2b presents the TGA of the 50 nm Fe powders. It could be inferred that the onset for oxidation of the powders was approximately 370 °C, which hinted at the effectiveness of the original oxide surface layer in preventing further oxidation.

The saturation magnetization (*M*s) values measured for the Fe powders reveal *M*s = 175 Am^2^/kg (50 nm), *M*s = 186 Am^2^/kg (1 µm) and *M*s = 187 Am^2^/kg (11 µm). These values are consistent with the Fe wt% extracted from XRD given that the theoretical saturation magnetization value for pure Fe is *M*s = 220 Am^2^/kg [27], except for the value of the 11 µm powder, which suggests that it might be partially oxidized as well. We speculate that the lower reactivity of the larger Fe particles leads to oxide layers that are amorphous, and therefore, undetectable for XRD.

Figure 3 shows the magnetization curves of Fe/SFO-oriented composite powders fabricated with 50 nm sized Fe particles with soft phase contents of 5 vol%, 10 vol% and 15 vol%, as well as the curves of the individual SFO and Fe (50 nm) phases. From the individual SFO and Fe phases curves, we can see that the hard phase (SFO) presents a coercive field of 324 kA/m, while the saturation magnetization (*M*s) value is 69 Am^2^/kg. The soft phase (Fe) shows coercivity at (*H*_C_) ~2 kA/m and *M*s = 186 Am^2^/kg. This value is lower than expected for Fe (~220 Am^2^/kg) [28], which is due to the fact that these Fe nanoparticles are protected by a Fe_3_Si layer at their surface, as indicated by the supplier, which lowers *M*s. The presence of this layer excludes the hypothesis of exchange-coupling between Fe and SFO powders in the mixtures. Regarding the nanocomposites, for the sample with 5 vol% Fe (red curve), *M*s = 75 Am^2^/kg and *H*_C_ = 312 kA/m. In the sample with 10 vol% Fe (green curve), *H*_C_ = 290 kA/m and *M*s = 84 Am^2^/kg. Finally, the sample with 15 vol% Fe (blue curve) presents *H*_C_ = 264 kA/m and *M*s = 91 Am^2^/kg. As expected, in hard-soft composites, coercivity decreases, while *M*s increases with increasing soft content. Based on the steep drop in *H*_C_ for 15 vol%, we consider 10 vol% Fe the sample that presents the most competitive compromise in magnetic performance (based on *H*_C_ and *M*s).

It is important to note that the S shape observed in the second quadrant of the demagnetization curve of the composites (more evident for the 15 vol% sample but displayed by the other two as well), as detailed in the inset of Figure 3, strongly hints at an absence of exchange-coupling between hard and soft magnetic phases [11,12,15,20]. Composites fabricated with 1 µm and 11 µm diameter Fe particles present similar trends (not shown). Table 1 summarizes the magnetic properties of the bonded magnets studied.

Remanent magnetization (*M*_R_) presents an interesting behavior, as shown in Figure 4. In a multiphase magnetic material, in the absence of coupling, remanence is an additive property [28], the value of which can be calculated using the expression:(1)$${\left[M_{r}\right]}_{exp}=M_{r,\;hard}*wt{\%}_{hard}+M_{r,\;soft}*wt{\%}_{soft}$$

Using the *M*_R_ values for Fe and SFO measured in Figure 2, and given the assumption that no exchange-coupling occurs in these composite powders, Figure 3 shows the expected (calculated) *M*_R_ values together with the values experimentally measured (extracted from Figure 3). By first analyzing the case of non-oriented powders, it can be observed that the calculated and measured values practically coincide for 50 nm Fe particles, and slight deviations are measured for the 1 µm Fe size. We attribute these fluctuations/deviations to the random arrangement of non-oriented particles.

A very different scenario occurs for magnetically oriented powders. As can be observed, for all Fe contents and 50 nm and 1 µm particle sizes, the measured remanence is larger than the expected calculated value. This evidence excludes the hypothesis of total decoupling between hard and soft phases and suggests a magnetizing-type coupling that is only activated in the magnetically oriented state. The plausible reasons for this observation will be discussed later in the manuscript.

### 3.2. Injection-Molded Hard-Soft Composite Magnets

In order to investigate the potential of these composite powders as dense magnets, anisotropic (magnetically aligned) injection-molded permanent magnets were fabricated at the Max Baermann GmbH pilot production line. Fe content of 10 vol% was selected, and three different types of injection-molded permanent magnets were fabricated using the three Fe powders presented above with different particle sizes.

Figure 5 shows the demagnetization curves, measured in a closed loop in a permagraph instrument, for the three Fe particle sizes and for a single soft phase content (10 vol%), as well as a reference 100% ferrite sample. Figure 5a shows that coercivity decreases and the squareness of the demagnetization curve is significantly affected. While squareness is mainly lost, it is important to remark that, in contrast with the VSM curves measured in Figure 3 in powders, the smooth shapes of the demagnetization curves in Figure 4 are indicative of a system behaving as a single magnetic phase, suggesting an interparticle coupling between two phases.

In Figure 5b, we observe that for pure SFO (black line), remanent polarization *J*_R_ = 0.248 T, for the composite with 50 nm Fe particles, *J*_R_ = 0.255 T, for 1 µm Fe particles *J*_R_ = 0.255 T and lastly for 11 µm, *J*_R_ = 0.250 T. As for the powders, while a linear combination of hard and soft remanences should lead to a ~10% decrease in remanence in the composite with respect to the pure ferrite magnets, we measure a 2.4% increase in remanence, with respect to the pure ferrite magnet, for 50 nm and 1 µm Fe particle sizes and for a 10 vol% soft content; while a 0.8% increase is measured for 11 µm particles. Hence, we observe an anomalous non-monotonous variation in *M*_R_ and *H*_C_ with the particle size increase.

Table 2 summarizes the magnetic properties of the bonded magnets studied.

The porosity of the bonded magnets was calculated by measuring the density of the samples and using the theoretical densities. The pure ferrite magnet presents the lowest porosity 3.2%, while the composite magnets have porosities between 6–6.2%. Given that volume magnetization *M* depends on porosity *p* according to the formula *M* = (1 − *p*)*M*_S_, this entails that the increase in *J* observed cannot be explained by porosity changes.

It is also worth noting that the injection molding process is carried out at 250 °C, which is below the temperature at which the Fe powders oxidize, according to Figure 2b, and therefore, we expect no oxidation of the soft phase.

Figure 6 shows the SEM characterization of the surface of the injection-molded magnet made with 10 vol% 11 µm Fe particles. The size and morphology of both phases in the system SFO/Fe can be distinguished, where the smaller SFO particles embedded within the polymer form a percolated matrix that surrounds the larger Fe particles, which seem to be isolated. Figure 6b in particular clearly shows the presence of a void around the Fe particle in the center of the micrograph that prevents it from being in direct contact with the surrounding SFO matrix. Although the reasons behind the formation of this microstructure have not been investigated, we speculate that the fluid dynamics during the process of polymer wetting may be affected by the presence of the significantly larger Fe particles. A crucial consequence of this absence of direct contact is that we can again discard that the increase in remanence is due to effective exchange-coupling at the interface [15]. This observation emphasizes the surprising single-phase behavior of the demagnetization curve of the composite magnets and the increase in remanence observed.

### 3.3. Micromagnetic Simulations

A micromagnetic study of magnetization reversal in SFO/Fe samples was performed using an approach specifically developed for modeling the magnetization distribution in nanocomposites. The details of this simulation technique can be found in [24,25]. In all presented simulations, a cubical modeling volume with sides measuring 200 nm was discretized into 400,000 mesh elements, each sized about 3 nm. This high-performance calculation not only allows us to recover the details of magnetization distribution inside the crystallites, but also enables us to study a significant number of different crystallites, which is important for investigating magnetic interactions between them. All modeled samples have 20 vol% porosity. Four standard contributions to the total micromagnetic energy are taken into account: external magnetic field, anisotropy, exchange coupling and magnetodipolar interaction energies. Periodic boundary conditions are used.

We performed simulations changing the particle diameter between 15 to 60 nm and for three soft phase concentrations: 5%, 10% and 15% (volume fractions of magnetic material). The exchange coupling between crystallites was set to zero and anisotropy axes were oriented in the initial direction of the magnetic field. For every parameter set, the magnetization reversal of the composite was modeled and the corresponding demagnetization curves were calculated. In this manner, remanence was extracted from every curve and is presented in Figure 7 as a function of the soft phase concentration for each crystallite size. In all cases, the calculated remanence decreases with both increasing soft content and increasing particle size, as expected in hard soft composites [3,4,29,30]. The larger the soft particle size, the steeper the decrease in remanence.

As stated above, our micromagnetic approach allows us to present the evolution of the magnetization distribution of individual iron crystallites in the sample. The spin configuration of Fe nanoparticles of diameters between 15–50 nm was simulated and the results are shown in Figure 8. It can be observed that Fe particles only behave as magnetically single-domain for diameter D = 15 nm. For larger diameters, such as the particles used in the composites fabricated here, a vortex configuration forms, with the external spins (those closer to the surface of the particle) forming a closed circular loop while the central spins are aligned, as if an aligned magnetic rod was located at the center of the Fe particles. Figure 8b portrays this by showing an augmented section of the spins inside a 50 nm Fe particle. These calculations only qualitatively agree with the theoretical threshold between single and multidomain regimes defined by the coherent size (around 24 nm) but also by the domain wall length (around 65 nm) [25] and they illustrate the magnetic vortex/multidomain structure of Fe particles even in the nanodomains.

It has been demonstrated, by the shape of the demagnetization curves of the powders and the micrographs of the magnets showing voids around Fe, that no exchange-coupling takes place between SFO and Fe particles. Under this circumstance, the increase in remanence experimentally measured for the composite samples (in both powder and injection molded magnet forms) can only be explained by a certain degree of alignment of the spins of the soft phase with the magnetization of the hard, which happens even if Fe particles are in a multidomain state.

Based on the difference between measured and calculated (using expression 1) remanence in the oriented powders (~10% on average) and the *M*_R_ of Fe and SFO, it can be inferred that the fraction of Fe spins that actually aligned with the hard phase at remanence, and thus contribute to the overall increase in the remanence of the magnet, is approximately 4%. Looking at the spin configuration in Figure 7, we suggest that a plausible explanation is that the internal spins of the vortex structure are aligned with the internal field created by the hard SFO phase; i.e., due to the dipolar interaction between the hard and soft phase. The self-demagnetizing field in the Fe particles, proportional to the Ms of Fe, easily overcomes the internal field created by SFO, especially near the particle surface due to the minimization of the magnetostatic energy, which makes the spins of the Fe particle circularly curl to minimize the stray fields. However, the internal spins in the vortex structure are subjected to far inferior self-demagnetizing fields and they are, therefore, more likely to align with the hard particles. This alignment will be parallel or antiparallel depending on the geometric distribution of the field lines inside the magnet, which in turn depends on the distance and geometric arrangement of SFO and Fe particles. It is nevertheless safe to assume that, given the parallel alignment of the magnetization of all SFO particles inside the magnetically oriented bonded magnet, the internal magnetic fields will lead to a net alignment of the soft spins in the direction parallel to the magnetization of the hard. This mechanism is consistent with the small (~4%) fraction of Fe spins that are estimated to be aligned in the magnet and the fact that the remanence increase, with respect to the theoretically expected, is observed irrespective of the Fe particle size.

## 4. Conclusions

The magnetic properties and the microstructure of SrFe_12_O_19_/Fe hard-soft composites, in powders and injection-molded magnet form, have been studied as a function of soft phase content -between 5–15 vol%- and soft particle size -between 50 nm–11 µm. While coercivity decreases with soft phase concentration, as expected in hard-soft composites, a remanence that is larger than expected is measured in both oriented powders and oriented-bonded magnets. In fact, the hard-soft composite injection-molded magnets present a 2.4% increase in remanence with respect to identically prepared pure ferrite magnets, for all particle sizes explored. The lack of exchange-coupling between hard and soft phases, evidenced by the absence of direct contact between SFO and Fe particles seen in the microstructure of the magnets and the shape of the demagnetization curves, points at dipolar interactions as the cause for the remanence increase observed. The micromagnetic simulations performed reveal that a vortex spin configuration can form in spherical Fe soft particles with diameters above 15 nm. We suggest that the spins at the core of the vortex align with the hard phase, explaining the observation and the fact that it occurs for all particle sizes studied and only when the particles are magnetically oriented. These results open pathways to improving the remanence in hard-soft ferrite-based composites in the absence of exchange-coupling, which would be of great interest as the strict requirements associated with effective exchange-coupled would not have to be met, which in turn enhances the applicability of the method at an industrial level. This has ramifications as well in the ultimate development of hard-soft permanent magnets with enhanced performance, of any composition.

## Figures and Tables

**Figure 1 nanomaterials-13-02097-f001:**
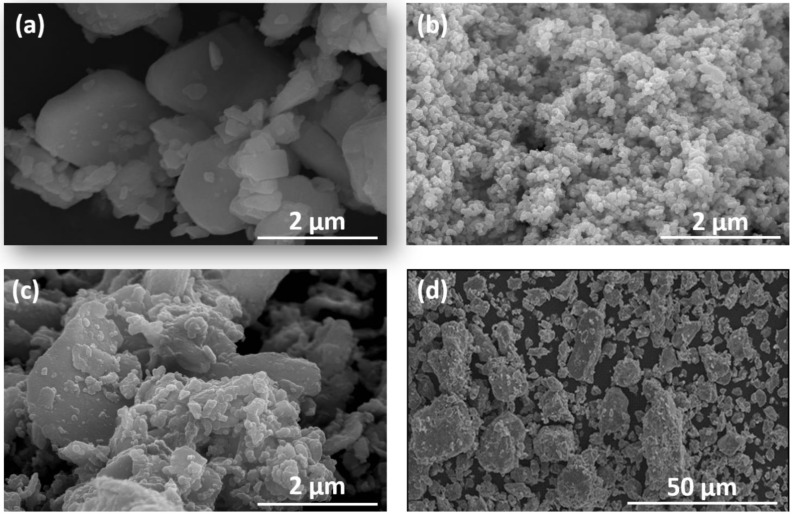
SEM micrographs showing particle size and morphology of the starting (**a**) SFO and Fe powders with (**b**) 50 nm, (**c**) 1 µm and (**d**) 11 µm average particle sizes.

**Figure 2 nanomaterials-13-02097-f002:**
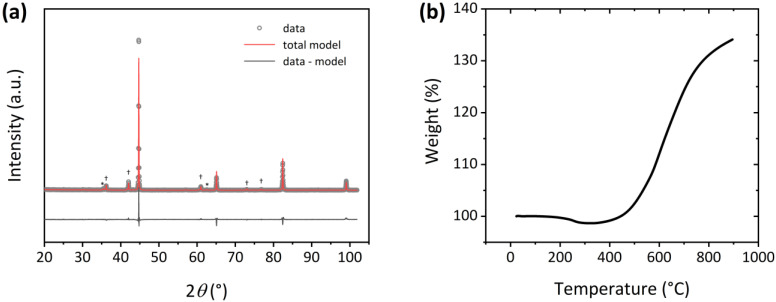
(**a**) XRD pattern and corresponding Rietveld refinement for the 50 nm Fe powder. † and * denote diffraction maxima from FeO and Fe_3_O_4_ respectively. (**b**) TGA of the 50 nm Fe powder performed in air heating between RT and 900 °C.

**Figure 3 nanomaterials-13-02097-f003:**
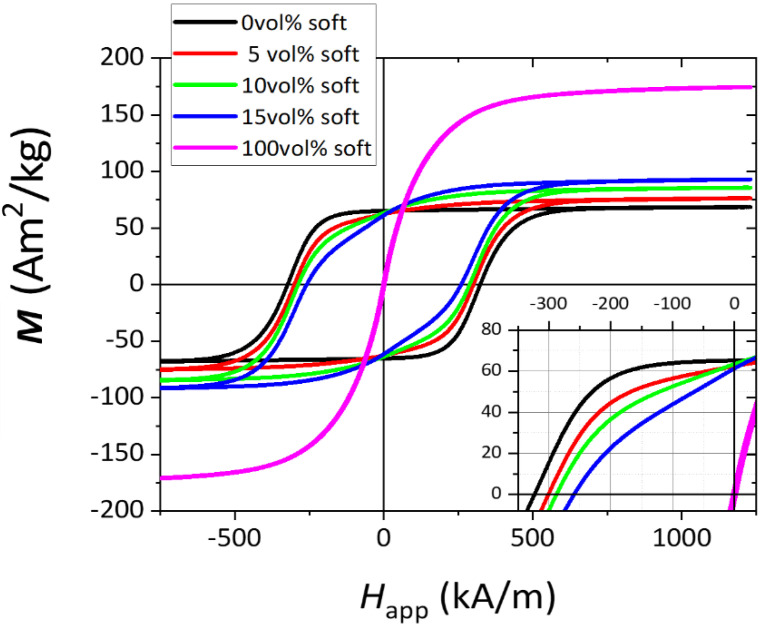
Magnetization vs. applied field curves of SFO/Fe composite powders using 50 nm Fe particles and for different soft phase concentrations, including the curves of the individual SFO and Fe phases.

**Figure 4 nanomaterials-13-02097-f004:**
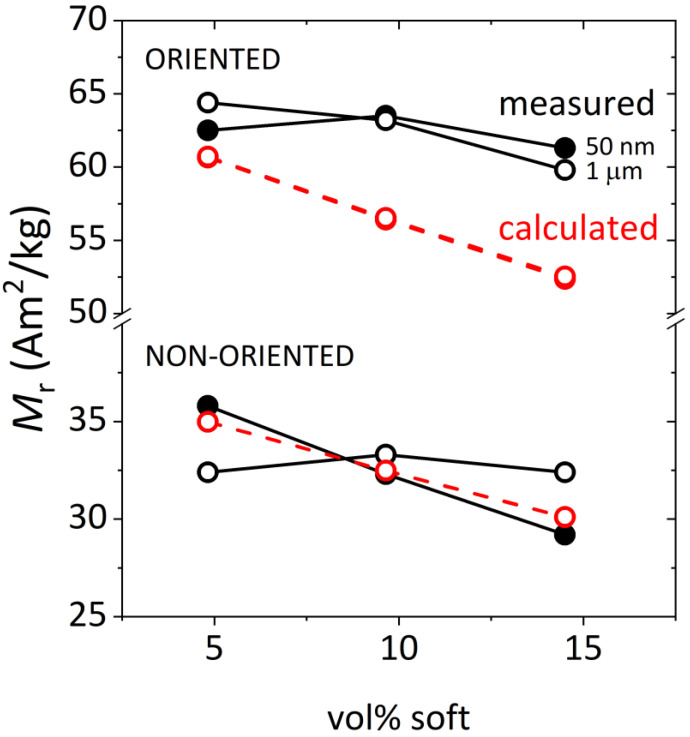
Remanence (*M*_R_) as a function of Fe content. The graph shows two groups of three curves, corresponding to the measured values of the composites fabricated with 50 nm and 1 µm Fe powders and the values calculated by the linear combination of the *M*_R_ values of Fe and SFO individual phases, for both oriented and non-oriented powders.

**Figure 5 nanomaterials-13-02097-f005:**
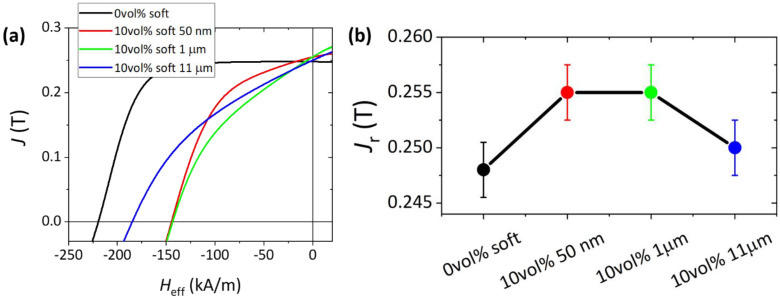
(**a**) Magnetic polarization *J* as a function of the applied magnetic field *H*eff of injection-molded composite magnets with 10 vol% Fe content for three different Fe particle sizes. (**b**) Remanence values for the four samples as a function of particle size.

**Figure 6 nanomaterials-13-02097-f006:**
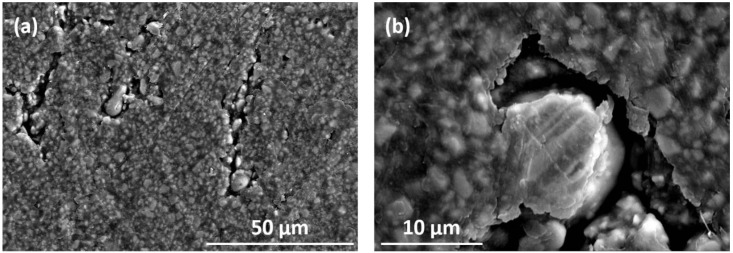
SEM images taken on the surface of the magnet at (**a**) ×1000 and (**b**) ×6000 magnifications showing the microstructure of injection-molded magnets with 10 vol% Fe particles of 11 µm.

**Figure 7 nanomaterials-13-02097-f007:**
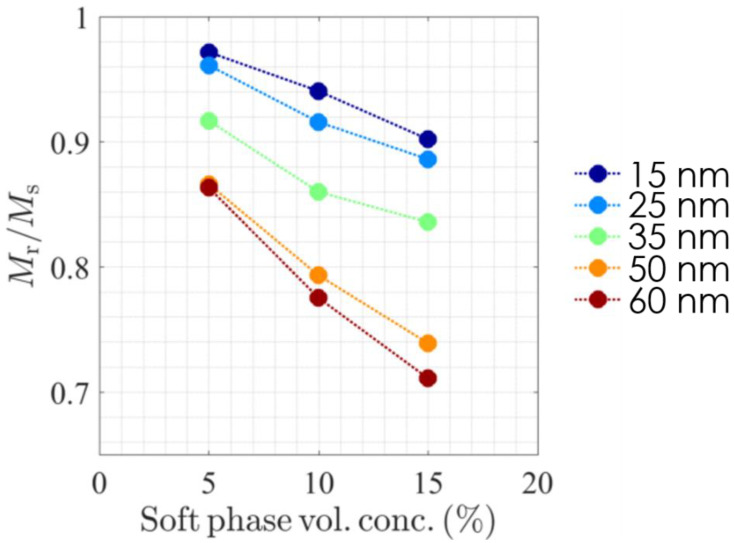
Micromagnetically calculated values of remanence in SFO/Fe composites as a function of Fe content and for different Fe particle sizes.

**Figure 8 nanomaterials-13-02097-f008:**
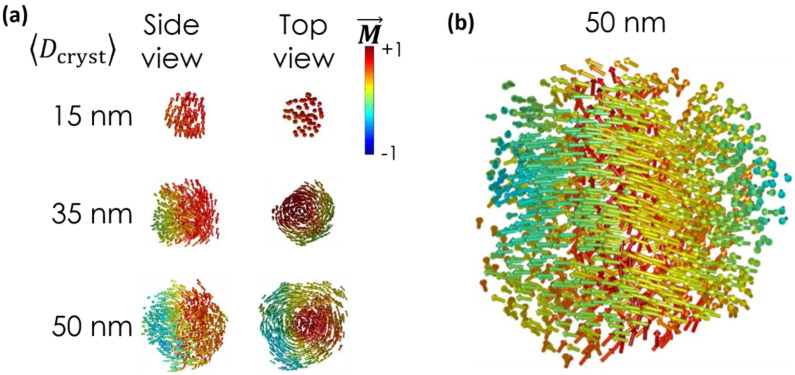
(**a**) Micromagnetically simulated images of the spin configuration of isolated Fe particles of different diameters between 15–50 nm. (**b**) Detail of the spin configuration of a 50 nm diameter Fe particle.

**Table 1 nanomaterials-13-02097-t001:** Magnetic parameters for the oriented powder samples.

Sample	*H*c (kA/m)	*M*_R_ (Am^2^/kg)	*M*_S_ (Am^2^/kg)	Type
SFO	323	65.1	68.7	Oriented Powder
SFO + 5 vol% Fe 50 nm	301	62.5	76.4	Oriented Powder
SFO + 10 vol% Fe 50 nm	287	63.5	86	Oriented Powder
SFO + 15 vol% Fe 50 nm	259	61.3	93.1	Oriented Powder
100% Fe 50 nm	3	3.7	174.6	Oriented Powder

**Table 2 nanomaterials-13-02097-t002:** Magnetic parameters and porosity of the injection moulded magnets.

Sample	*H*_C_ (kA/m)	*J* (T)	Type	Porosity (%)
SFO	219.6	0.248 ± 0.0025	Bonded Magnet	3.8
SFO + 10 vol% Fe 50 nm	144.8	0.255 ± 0.0025	Bonded Magnet	6.2
SFO + 10 vol% Fe 1 µm	143.6	0.255 ± 0.0025	Bonded Magnet	6.2
SFO + 10 vol% Fe 11 µm	184.7	0.25 ± 0.0025	Bonded Magnet	6

## Data Availability

The data is available on reasonable request from the corresponding author.

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
