# Peer review of "Remanence Increase in SrFe12O19/Fe Exchange-Decoupled Hard-Soft Composite Magnets Owing to Dipolar Interactions"

_nanomaterials, 2023, doi:10.3390/nano13142097_

Round 1

Reviewer 1 Report

Dear Editor,

The authors presented their studies on the remanence of hard-soft SrFe12O19/Fe composites in terms of use them as permanent magnets. In order to improve the manuscript, I have the following suggestions for the authors.

Please, describe the method of obtaining a magnetically oriented "powder" sample in more detail. What is the applied magnetic field during the orientation process. Does the polymer used isolate the individual particles or clusters of them.

In Figure 5, it is not clear whether the images are on the surface or on a cross-section of the magnet.

Please, show in the table the value of magnetic characteristics for all samples.

Reviewer 2 Report

There has been considerable interest in generating rare-earth free permanent magnets with greater MH product, and efforts such as these are important in unlocking the answer to that technological challenge. That said, this work can contribute to the discussion but major issues need to be addressed first. 1) It is important to know the magnetic porosity, not only for determining the theoretical saturation magnetization of the sum of the components but also for calculating the dipolar effects since any break in the effective magnetic circuit significantly changes the field lines. 2) It is important to know the oxidation of the iron. All Fe particles oxidize on the surface and that will likely change with processing. The effect will certainly vary with particle size and be most significant for the nanoparticles. Since all Fe oxides have significantly smaller magnetization than that of elemental Fe, and at best comparable to that of SRO, it is again important to both understanding the measured value as well as generating the appropriate model. 3) The sample size and shape are not given. Sample shape plays an important role in the demagnetizing field and thus the remanent magnetization.  This should also be included in the simulations. These are very important issues that are critical to the paper. If they can be addressed, I encourage the authors to resubmit, but I cannot recommend at this time.

Reviewer 3 Report

The manuscript addresses the Mr increase in hard soft magnetic composites. The content is clearly presented and can be followed with no issues. 

Technical comments provided below:

- Although not commonly recognized, the figure of merit for permanent magnets will depend on the application, so that BHmax in "one" of the figures of merit, not "the" figure of merit. Example: in certain motor topologies coercivity is the figure the merit because the motor operation, although dependent on the magnetic flux from the magnet, creates the main portion of torque based on other physical phenomena.

- Shouldn't Eq. (1) be vol% instead of wt%?

- Main point of clarification is Figure 4:

   - Figure 4b: are data points representing single magnets or this is an average of many produced parts? Please explain.

   - Either way, error bars must be added to Figure 4b. With this addition, could the conclusions of the authors be changed? Notice that in case an error of plus/minus 1% is present, the dataset  will be in the same "corridor", and the 2.4% difference claimed is not applicable anymore. 

Round 2

Reviewer 3 Report

Publish as it is.